# Attenuated Chromosome Oscillation as a Cause of Chromosomal Instability in Cancer Cells

**DOI:** 10.3390/cancers13184531

**Published:** 2021-09-09

**Authors:** Kenji Iemura, Yujiro Yoshizaki, Kinue Kuniyasu, Kozo Tanaka

**Affiliations:** Department of Molecular Oncology, Institute of Development, Aging and Cancer, Tohoku University, Seiryo-machi, Aoba-ku, Sendai 980-8575, Japan; kenji.iemura.a6@tohoku.ac.jp (K.I.); yoshizaki.yujiro.q3@dc.tohoku.ac.jp (Y.Y.); kinue.kuniyasu.b1@tohoku.ac.jp (K.K.)

**Keywords:** chromosomal instability, cancer, chromosome oscillation, kinetochore-microtubule attachment, Aurora kinase, aneuploidy, Hec1

## Abstract

**Simple Summary:**

Chromosomal instability (CIN), a condition in which chromosome missegregation occurs at high rates, is widely seen in cancer cells. Causes of CIN in cancer cells are not fully understood. A recent report suggests that chromosome oscillation, an iterative chromosome motion typically seen in metaphase around the spindle equator, is attenuated in cancer cells, and is associated with CIN. Chromosome oscillation promotes the correction of erroneous kinetochore-microtubule attachments through phosphorylation of Hec1, a kinetochore protein that binds to microtubules, by Aurora A kinase residing on the spindle. In this review, we focused on this unappreciated link between chromosome oscillation and CIN.

**Abstract:**

Chromosomal instability (CIN) is commonly seen in cancer cells, and related to tumor progression and poor prognosis. Among the causes of CIN, insufficient correction of erroneous kinetochore (KT)-microtubule (MT) attachments plays pivotal roles in various situations. In this review, we focused on the previously unappreciated role of chromosome oscillation in the correction of erroneous KT-MT attachments, and its relevance to the etiology of CIN. First, we provided an overview of the error correction mechanisms for KT-MT attachments, especially the role of Aurora kinases in error correction by phosphorylating Hec1, which connects MT to KT. Next, we explained chromosome oscillation and its underlying mechanisms. Then we introduced how chromosome oscillation is involved in the error correction of KT-MT attachments, based on recent findings. Chromosome oscillation has been shown to promote Hec1 phosphorylation by Aurora A which localizes to the spindle. Finally, we discussed the link between attenuated chromosome oscillation and CIN in cancer cells. This link underscores the role of chromosome dynamics in mitotic fidelity, and the mutual relationship between defective chromosome dynamics and CIN in cancer cells that can be a target for cancer therapy.

## 1. Introduction

Most cancer cells have an abnormal number and structure of chromosomes [1,2]. Gain or loss of entire chromosomes is called (whole chromosome) aneuploidy, while amplification or loss of parts of chromosomes is called structural (or segmental) aneuploidy [3,4,5]. Aneuploidy is caused by chromosome missegregation during mitosis, which is derived not only from mitotic defects, but also from defects in interphase such as replication stress [6]. In many cases, aneuploid cancer cells exhibit increased rates of chromosome missegregation, which is called chromosomal instability (CIN) [7,8,9]. CIN is a cause of intratumor heterogeneity, and related to cancer progression and poor prognosis, including metastasis and drug resistance [10,11,12,13]. Various paths to CIN have been revealed, although a whole picture of the etiology of CIN in cancer has not yet been clarified [7,14]. Proper attachment of a kinetochore (KT), a proteinous structure formed at the centromeric region on a chromosome, to spindle microtubules (MTs) is a prerequisite for faithful chromosome segregation. MTs are cylindrical assembly of protofilaments formed by heterodimers of α and β tubulin. This structure stochastically repeats elongation (by polymerization of tubulin heterodimers) and shrinkage (by depolymerization of tubulin heterodimers), a phenomenon known as dynamic instability that facilitates attachment to KTs [15]. Failure to achieve proper KT-MT attachments is involved in the generation of CIN [16,17]. Dysfunction of the mechanisms to resolve erroneous KT-MT attachments (error correction mechanisms) plays a particularly crucial role in the emergence of CIN [18,19]. Mitosis is a dynamic process where KT-MT attachments are regulated both temporally and spatially, and chromosome dynamics is closely linked to the formation of these attachments [20,21,22,23,24]. For example, in prometaphase, chromosome alignment at the spindle equator (congression) is required for the establishment of proper KT-MT attachments and faithful chromosome segregation [24,25]. In metaphase, chromosomes are aligned at the spindle equator, forming the so-called metaphase plate. These aligned chromosomes are not just staying at the metaphase plate, but moving around the spindle equator periodically, a phenomenon known as chromosome oscillation [26,27]. In this review, we first provide an overview of the error correction mechanisms of KT-MT attachments and the features and mechanisms of chromosome oscillation. We then introduce a previously unappreciated link between chromosome oscillation and error correction mechanism, and also describe attenuated chromosome oscillation as a novel cause of CIN in cancer cells.

## 2. Correction Mechanisms of Erroneous KT-MT Attachments and Its Relevance to CIN

### 2.1. Overview of Chromosome Dynamics in Mitosis

Chromosome segregation is carried out on the spindle, comprising two spindle poles and MTs that connect spindle poles to KTs (see Figure 1). Mitosis is divided into five phases: prophase, prometaphase, metaphase, anaphase, and telophase (see Figures 3 and 4). Following prophase, when chromatin condenses into well-defined chromosomes, prometaphase starts with nuclear envelope breakdown (NEBD). During prometaphase, spindle is formed while MTs attach to KTs. In metaphase, chromosomes align to the middle of the spindle, forming the metaphase plate. Anaphase onset is marked by the synchronous separation of all sister chromatids, and the separated chromosomes move towards the poles. In telophase, nuclear envelope reforms around the clustered chromosomes, and chromatin decondenses. These are common processes among eukaryotic cells, although there are many variations between species. In animal cells, spindle poles are defined by centrosomes that act as MT-organizing centers (MTOCs), while equivalent structures in fungi are called spindle pole bodies. Higher plant cells and the oocytes in many animal species do not have centrosomes. NEBD does not occur in many lower eukaryotes such as yeast, and they undergo a closed mitosis by forming the spindle in the nucleus. The number of MTs attaching to KTs also differ between species: one for a KT in budding yeast (*Saccharomyces cerevisiae*) [28,29], 2–4 for that in fission yeast (*Schizosaccharomyces pombe*) [30], and 20–30 in animal cells (kinetochore-fiber or K-fiber) [31,32]. On the other hand, chromosomes in *C. elegans* contain multiple centromeres along chromosome arms, which is called holocentric [33]. In the following sections, we mainly describe the mechanisms in mammalian cells.

### 2.2. Proper and Erroneous KT-MT Attachments

For equal chromosome segregation to daughter cells, a KT pair on sister chromatids has to be attached to MTs from opposite spindle poles, which is called bi-orientation, or amphitelic attachment (Figure 1A [16,34]. After NEBD, KTs initially attach to the lateral surface of MTs (lateral attachment), and are transported to the spindle surface by a minus end-directed motor dynein [35,36,37,38,39,40,41]. Then chromosomes are transported toward the spindle equator along spindle MTs by a process called chromosome congression, which involves the cooperative actions of two plus end-directed motor proteins, CENP-E on KTs and Kid on chromosome arms [20,21,42,43,44,45]. Lateral attachment is then converted to end-on attachment, in which MT ends attach to KTs, forming K-fiber [22,46,47]. In the process of establishment of bi-orientation, three types of erroneous KT-MT attachments can arise (Figure 1B–D) [16]. One is monotelic attachment, a situation where only one of the sister KTs attaches to MTs. The second is syntelic attachment, in which both sister KTs attach to MTs from the same spindle pole. The third is merotelic attachment, where a single KT attaches to MTs from both spindle poles. The back-to-back geometry of sister KTs facilitates the formation of bi-orientation, by making a sister KT face one spindle pole when the other sister KT attaches to MTs from the opposite spindle pole [48,49,50]. Chromosome congression through lateral attachments also promotes bi-orientation establishment by placing KTs at the spindle equator, where MTs from both spindle poles exist at comparable density [22,51]. However, erroneous KT-MT attachments are still formed frequently, partly due to the incomplete separation of centrosomes at nuclear envelope breakdown, which become spindle poles in mitosis [35,51,52,53]. KT expansion in early mitosis [23,54,55], which increases the chance for MT attachment, also contributes to the increased risk of erroneous attachment formation. To resolve these erroneous attachments, there are mechanisms, referred to as error correction mechanisms, which ensure mitotic fidelity [17,56].

### 2.3. Error Correction Mechanisms of KT-MT Attachments

Error correction of KT-MT attachments works by destabilizing erroneous attachments while stabilizing correct ones. As a premise to enable the error correction, KT-MT attachments have to be adequately dynamic; not too unstable, but also not too stable [57,58]. Error correction mainly occurs in prometaphase, when MTs initially attach to KTs in a stochastic manner through the process of dynamic instability [40,59,60,61]. It was reported that KT-MT attachments in prometaphase are less stable than that in metaphase, making it more suitable for efficient error correction [62,63]. The stability of KT-MT attachment is defined by turnover rates of MTs on KTs and the stability of MTs forming K-fibers. Among the KT components, the KMN network is responsible for connecting KT to MT [64]. Locating at the outer part of KT (outer plate), this network is comprised of the Knl1 complex, the Mis12 complex, and the Ndc80 complex. The Ndc80 complex is a heterotetramer composed of Hec1 (Ndc80), Nuf2, Spc24, and Spc25 (Figure 2A). Human Hec1 (highly expressed in cancer) was originally identified as an Rb-binding protein that is highly expressed in several cancers [65,66]. The globular calponin-homology domain formed by Hec1 and Nuf2 directly binds to MTs, playing a major role in the attachment to MTs [67,68]. The disordered N-terminal region of Hec1 (Hec1 tail) has nine phosphorylation sites for Aurora kinases, and their phosphorylation reduces its affinity to MTs, which is the main mechanism for error correction (Figure 2B) [69,70,71]. This Hec1 phosphorylation is counteracted by phosphatases, which stabilizes KT-MT attachment [69,72,73,74,75,76]. Other components of the KMN network are also phosphorylated by Aurora B and cooperatively regulate the affinity to MTs [77]. Stability of MTs forming K-fiber is regulated by various MT-binding proteins. Among them, two kinesin-13 family motor proteins that promote MT depolymerization, Kif2b, and MCAK (Kif2c), and destabilize MTs, forming K-fibers specifically in prometaphase and metaphase, respectively [62].

Two Aurora kinases, Aurora A and B, are critical for proper chromosome alignment and maintenance of mitotic fidelity [78]. While Aurora A localizes to spindle poles and the spindle, Aurora B localizes to the inner centromere in early mitosis, spindle midzone in anaphase, and midbody in telophase and cytokinesis (Figure 3) as a component of the chromosome passenger complex, comprising Aurora B, INCENP, Survivin, and Borealin [71,79]. Aurora B is a key player for error correction that destabilizes erroneous KT-MT attachments, which is explained by the “spatial separation model” [80,81,82]. When bi-orientation is established, sister KTs are under tension due to MTs pulling towards opposite spindle poles, causing an increase in inter-KT distance. The resulting separation of the outer plate from inner centromere leads to the reduction of Hec1 tail phosphorylation by Aurora B, thereby increasing the affinity to MTs and stabilizes KT-MT attachments [77]. In contrast, sister KTs forming syntelic attachments are not under tension, and thus Hec1 phosphorylation continues. This facilitates KT detachment, which allows another chance for MT attachment. In the case of merotelic attachment, KTs are supposed to be elongated towards the inner centromere through MT pulling force to the opposite spindle pole. It is suggested that Aurora B phosphorylates Hec1 on the elongated KT portions, thus destabilizing merotelic attachment [83,84,85,86]. This spatial separation model, which is based on Aurora B localization at the inner centromere, can explain tension-dependent error correction. However, recent reports suggest the possibility that MT binding of Aurora B through INCENP is critical for error correction by selective destabilization of erroneous attachments [87,88]. A recent report that exploited optogenetics to manipulate Aurora B at individual KTs showed that Aurora B rather promotes MT release under high tension while depolymerizes MT under low tension [89]. Other Aurora B substrates, such as MCAK [90,91], CENP-E [92], the SKA complex [93], and HURP [94,95], also contribute to the establishment and stabilization of bi-orientation.

Recently, it was reported that Aurora A also plays a role in the correction of erroneous KT-MT attachments [96,97]. Aurora A, which plays a role in spindle assembly, mainly localizes to spindle poles and the spindle near spindle poles (Figure 3). Aurora A and B share many substrates, and their specificity is mainly determined by the proximity of the kinases to the respective substrates [98]. In prometaphase when chromosomes are near spindle poles, Hec1 on the KTs is phosphorylated and KT-MT attachments are destabilized. This resolves the erroneous attachments formed in early mitosis, especially syntelic attachment that tends to be formed when chromosomes are closest to the spindle pole.

Mps1 is a kinase that plays a crucial role in the spindle assembly checkpoint (SAC) and also functions in error correction through its interplay with Aurora B [99,100,101,102,103,104]. It was reported that Chk1, which phosphorylates and activates Aurora B [105], is also involved in error correction together with Mps1 [106]. A recent report suggests that Cdk1, a main kinase important for mitotic progression, phosphorylates the Hec1 tail, which may be involved in error correction (*bioRxiv* doi:10.1101/2021.02.16.431549). As another mechanism of error correction, ch-TOG (Stu2 in budding yeast) is involved in the correction of erroneous attachments by stabilizing KT-MT attachment when tension is exerted [107,108].

### 2.4. CIN Caused by Insufficient Correction of Erroneous KT-MT Attachments

Erroneous KT-MT attachments are corrected by various mechanisms in a context-dependent manner, as mentioned in the previous section. For example, an unattached KT in a monotelically-attached sister KT pair is sensed by the SAC, which halts anaphase onset until both sister KTs attach to MTs [109]. Syntelically-attached KTs, which are under low tension, are also sensed by the SAC [110,111]. In contrast, merotelic attachment is not sensed by the SAC, because merotelically-attached KTs are attached to MTs and are under tension [25]. It is thus considered that merotelic attachment is a major cause of CIN, as chromosome segregation can occur in the presence of merotelic attachments if they are left uncorrected [86,112]. Even when uncorrected, merotelic attachments are resolved during spindle elongation in most cases [113]. However, remaining merotelic attachments are liable to form lagging chromosomes in anaphase, which are left behind around the spindle equator while other chromosomes are segregated to the respective spindle poles. Lagging chromosome is one of the typical patterns of chromosome missegregation. This can cause aneuploidization when they are ultimately segregated to the wrong side, or micronuclei formation when they are excluded from the main nuclei [114]. It is known that replication and repair of chromosomes in micronuclei are often defective [115,116,117,118], which results in structural chromosomal abnormalities including chromothripsis, a mutational phenomenon characterized by extensive genomic rearrangements of one or a few chromosome(s) [119,120]. Lagging chromosomes are also sometimes stuck in the cleavage furrow, resulting in structural chromosomal abnormalities due to DNA damage and cytokinesis failure through furrow regression [121]. Collectively, merotelic attachment is causative of CIN by generating both numerical and structural chromosomal abnormalities [122].

Merotelic attachments are increased either by increased formation or insufficient correction. A cause of increased formation of merotelic attachments is centrosome amplification, which is often observed in cancer cells [123,124]. To avoid multipolar spindle formation that results in catastrophic multipolar division [125], cells form pseudo-bipolar spindles by clustering excessive centrosomes (centrosome clustering) [126,127,128,129]. However, the process of centrosome clustering increases the chance of merotelic attachment formation [128,129]. A cause of insufficient correction of merotelic attachments is MT hyperstabilization, which hampers the destabilization of these attachments [57,62,130]. Another cause of inefficient error correction is reduced Aurora B activity. It was reported that Aurora B is enriched at misaligned centromeres in non-transformed cells, but not in aneuploid tumor cells [131]. A recent report suggests that heterochromatin protein 1 (HP1) binds to INCENP and augments Aurora B activity. In cancer cells, the HP1 binding to INCENP is reduced, which results in insufficient error correction due to reduced Aurora B activity [132].

## 3. Chromosome Oscillation

### 3.1. The Features of Chromosome Oscillation

On the metaphase plate, chromosomes, especially KTs, wobble around the spindle equator, which is called chromosome (or KT) oscillation (Figure 4) [26,27]. Chromosome oscillation is widely seen from yeast to human cells in mitosis and meiosis, but is not observed in some cases, such as in mitosis of insect cells, *Xenopus* egg extracts, and plant cells [133,134,135,136,137,138]. The main driving force of chromosome oscillation is the MT pulling force exerted by K-fibers. Although individual MTs within K-fiber behave independently, their dynamics is coordinated by MT-binding proteins, and collectively drive iterative chromosome motion, which is also known as directional instability [139,140,141]. This directional instability is seen not only in metaphase, but throughout mitosis, and also seen on monopolar spindles [142]. Chromosome oscillation declines as cells progress toward anaphase [143,144], reflecting the stabilization of KT-MT attachments for chromosome segregation.

### 3.2. The Mechanisms of Chromosome Oscillation

In RPE-1 cells, a non-transformed cell line, the duration of metaphase is around 10 min, while one round of chromosome oscillation spans 1 to 2 min [145,146,147]. Therefore, a pair of sister chromatids oscillates ~10 cycles during metaphase before segregation. Symmetrical chromosome motion during oscillation is derived from the tug of war of K-fibers attaching to sister KTs. In metaphase, directional instability of chromosomes changes the distance between a KT and a spindle pole, the distance between the centroid of sister KTs and the spindle equator, and the distance between sister KTs [148,149]. When one of the sister KTs is pulled toward a spindle pole by a shrinking K-fiber (called leading KT), it causes a stretching of the elastic inter-KT region, which is comprised of the inner centromere and cohesin bond [150]. Another sister KT (called trailing KT) follows the leading KT, pulled by the cohesin bond. Pulling force on the leading KT is increasingly counteracted by other forces (see below) as it approaches a spindle pole. Consequently, directional instability is triggered, which switches the role of the KT from leading to trailing. During the process, inter-KT distance reduces until the new leading KT stretches the inter-KT region. Therefore, the inter-KT stretch occurs twice the frequency of the oscillation of the centroid of sister KTs [151]. An important molecule that regulates chromosome oscillation is a motor protein called Kif18A [152,153]. Kif18A is a kinesin-8 family of plus end-directed motor protein, and accumulates at plus ends of MTs in K-fibers. At plus ends, Kif18A suppresses the MT polymerization, thus restricting chromosome oscillation [152]. Longer MTs accumulate more Kif18A at the plus ends, as the protein moves along MTs [154]. Therefore, polymerization is suppressed preferentially on longer MTs, limiting the range of chromosome oscillation. Kid and Kif4A, collectively called chromokinesins, are other motor proteins involved in chromosome oscillation, which localize to chromosome arms [153,155]. Kid is a plus end-directed motor belonging to the kinesin-10 family, which pushes chromosomes to the spindle equator along spindle MTs, known as polar ejection force or polar wind [156,157]. As the density of spindle MTs is higher in the vicinity of spindle poles, polar ejection force increase as chromosomes are closer to spindle poles, restricting the amplitude of chromosome oscillation [153,158]. Polar ejection force also contributes to chromosome oscillation on monopolar spindles as an opposing force against the pulling force by K-fibers, although the oscillatory motion is not symmetric as the one in metaphase [159]. A MT-associating protein NuSAP was reported to play a role in chromosome oscillation by regulating Kid-generated polar ejection force [160]. Kif4A is a kinesin-4 family plus end-directed motor protein that also plays a role in chromosome oscillation by regulating MT dynamics. At spindle poles, a kinesin-13 motor called Kif2A engages in the depolymerization of MT minus ends in K-fibers, called MT flux, which facilitates the pulling of chromosomes to spindle poles [161,162]. Recently, it was proposed that MT flux is driven by Kif4A on chromosome arms in coordination with Eg5, a kinesin-5 family, and Kif15, a kinesins-12 family motor protein, which slide antiparallel MTs [163]. Collectively, this induces Kif2A-dependent MT depolymerization. At KTs, proteins other than the Ndc80 complex, such as CENP-H, ch-TOG, and SKAP, are also involved in chromosome oscillation [146,164,165]. It was recently reported that K-fibers on sister KTs are connected by MTs that overlaps in the middle of the spindle, referred to as bridging MTs [166]. Bridging MTs contribute to the generation of tension between sister KTs and also chromosome congression and oscillation [167]. Bridging MTs branch from K-fibers via the augmin complex (*bioRxiv* doi:10.1101/2020.09.10.291740), and opposing bridging MTs are connected with PRC1 at the overlapping region [168]. It was suggested that bridging MTs associate with K-fibers, and the range of chromosome oscillation is determined by the length of the antiparallel overlaps of bridging MTs, which is regulated by Kif4A and Kif18A [167,169].

Chromosome oscillation is a complex process driven by various forces acting on chromosomes, and its underlying mechanism has been addressed by numerical models [26,141,170,171,172,173,174,175,176]. In these models, forces acting on a chromosome are considered in equilibrium at a given time (force balance model) [171,176]. Viscous resistance from cytoplasm is included as well as pulling force by K-fiber through Hec1, polar ejection force, and pulling force from sister KT through cohesin bonds (Figure 5). To enable oscillatory motion, several assumptions are integrated depending on the models, including force-dependent detachment kinetics of the Ndc80 complex, length and polymerization rate-dependent MT catastrophe frequency, position-dependent polar ejection force, and position-dependent activity gradient that weakens affinity of KT to MT [174,177,178]. These models help us to understand the relative contribution of each force on chromosome oscillation.

## 4. Chromosome Oscillation Plays a Role in Correction of Erroneous KT-MT Attachments

### 4.1. Hec1 Phosphorylation by Aurora A in Metaphase

As described above, Hec1 phosphorylation by Aurora kinases is reduced when bi-orientation is established and sister KTs are under tension. However, when observed carefully, it was found that in RPE-1 cells, serine 55 on Hec1 (Hec1-S55) is phosphorylated in a fraction of KTs in metaphase, preferentially at the periphery of the metaphase plate [147]. In contrast, Hec1-S55 phosphorylation in metaphase was barely detectable in HeLa cells, a cervical cancer-derived cell line. This Hec1 phosphorylation in metaphase was dependent on Aurora A, but not Aurora B. This was determined by specific inhibitors for the respective kinases, and metaphase-specific depletion of either kinase with the auxin-inducible protein degradation method [179,180,181]. Aurora A-dependent Hec1 phosphorylation in metaphase was also reported for serine 69 of Hec1 (Hec1-S69), which was observed on KTs throughout the metaphase plate, and attributed to a fraction of Aurora A in the inner centromere [182]. In contrast, Hec1-S55 phosphorylation was dependent on Aurora A localizing to the spindle, which was shown by reduction of the phosphorylation when spindle localization of Aurora A was inhibited [147]. This was achieved by replacing endogenous TPX2, a MT-binding protein that recruits Aurora A to the spindle, with TPX2 mutants unable to bind to Aurora A.

As described, Hec1 phosphorylation in prometaphase by Aurora A localizing around spindle poles was previously reported when KTs were closest to spindle poles [96,97]. The Hec1 phosphorylation on KTs near spindle poles resolve monotelic or syntelic attachments, which otherwise are stabilized by the tension exerted by KT pulling force via end-on-attached MTs and polar ejection force on chromosome arms [183]. However, it was not known whether Aurora A localizing near spindle poles plays a role in error correction in metaphase, when chromosomes are distant from spindle poles. In metaphase, Aurora A localization on the spindle increases upon mature K-fiber formation [147], which may extend the phosphorylation activity gradient closer to the metaphase plate.

### 4.2. Chromosome Oscillation Promotes Hec1 Phosphorylation by Aurora A

Amplitude of chromosome oscillation in HeLa cells was significantly smaller compared with that in RPE-1 cells [147]. This amplitude was increased when Kif18A was depleted by siRNA or treatment with a highly specific small molecule inhibitor. In this situation, the Hec1-S55 phosphorylation by Aurora A in HeLa cells was increased. On the other hand, when chromosome oscillation was suppressed in RPE-1 cells by reducing MT dynamics through treatment with taxol, a MT stabilizing agent, Hec1-S55 phosphorylation by Aurora A was reduced. These data suggest that chromosome oscillation expedites Hec1 phosphorylation by Aurora A. Considering that Aurora A distribution on the spindle is higher near spindle poles and KTs at the periphery of the metaphase plate are preferentially phosphorylated at Hec1-S55, chromosome oscillation facilitates Hec1-S55 phosphorylation by moving KTs closer to the area of higher Aurora A activity near spindle poles (Figure 6).

It is known that chromosome oscillation is significantly suppressed in cells expressing a Hec1 mutant in which all nine Aurora kinase-dependent phosphorylation sites were mutated to alanine [184]. This is probably due to suppression of K-fiber dynamics by hyperstabilized KT-MT attachments. It was also shown that chromosome oscillation is suppressed by inhibiting Aurora A, but not Aurora B [147,182]. These data indicate that Hec1 phosphorylation by Aurora A is required for robust chromosome oscillation. Analyzing Hec1 constructs in which respective phosphorylation sites were mutated, it was found that phosphorylation of Hec1-S55 and S69 cooperatively promotes chromosome oscillation [147,182]. Collectively, chromosome oscillation and Hec1 phosphorylation by Aurora A mutually promote each other.

### 4.3. Chromosome Oscillation Is Attenuated in CIN Cancer Cell Lines

Chromosome oscillation observed in cancer cell lines were significantly attenuated compared to non-transformed cell lines (Figure 6) [147]. Compared to CIN cancer cell lines (e.g., U2OS, HeLa, A549, DU145, and MCF-7 cells) which show high rates of chromosome missegregation, other cancer cell lines (e.g., HCT116, HCT-15, and DLD-1 cells, so-called “non-CIN” cell lines) show lower rates of chromosome missegregation. Interestingly, these non-CIN cell lines exhibit milder attenuation of chromosome oscillation compared with CIN cell lines, showing that the amplitude of chromosome oscillation is inversely correlated with the level of CIN.

The cause of attenuated chromosome oscillation in CIN cancer cell lines has not been specified yet. The amount of Aurora A on the spindle did not differ significantly depending on the CIN levels, excluding the possibility that difference in the spindle localization of Aurora A causes attenuation of chromosome oscillation in CIN cells [147]. It is known that KT-MT attachment stability is higher in CIN cells than those in non-transformed cells [62,130]. When activity of MCAK, a kinesin-13 family motor protein that destabilize MTs, was upregulated in HeLa cells, the amplitude of chromosome oscillation increased, suggesting that MT hyperstabilization is related to attenuated chromosome oscillation in CIN cancer cells. Another possibility is that the balance between opposing motor protein activities is altered in cancer cells. Expression of most of the mitotic motor proteins are upregulated in the majority of cancers (Figure 7). In particular, multiple types of cancer display elevated levels of Kif18A [185,186,187], which suppresses the amplitude of chromosome oscillation [152]. Recently, three papers reported that CIN or aneuploid cancer cells are vulnerable to Kif18A depletion, which causes spindle defects such as multipolar spindle formation [188,189,190]. It was suggested that increased rates of spindle MT polymerization in CIN cells confer an enhanced dependence on the role of Kif18A to limit MT growth [190,191]. One plausible idea is that Kif18A upregulation enables tumor cell survival through spindle assembly in prometaphase at the expense of CIN caused by attenuated chromosome oscillation in metaphase.

### 4.4. Chromosome Oscillation Facilitates Correction of Erroneous KT-MT Attachments

The amplitude of chromosome oscillation was inversely correlated not only with the CIN level, but also with the level of Hec1-S55 phosphorylation; cancer cell lines exhibit reduced Hec1-S55 phosphorylation depending on the CIN level [147]. This implies that Hec1 phosphorylation by Aurora A in metaphase, which is facilitated by chromosome oscillation, plays a role in the error correction of KT-MT attachments. Its dysfunction is related to CIN in cancer cell lines (Figure 6). This idea is supported by the fact that Aurora A depletion or inhibition in RPE-1 cells in metaphase led to increased chromosome missegregation. Chromosome oscillation enhancement via Kif18A inhibition also led to reduced chromosome missegregation, further supporting the idea that dysfunctions in chromosome oscillation is related to CIN. It was reported that Hec1 phosphorylation specifically affects KT attachments to polymerizing MTs [192]. It is postulated that when a leading KT approaches a spindle pole, Hec1 on the KT is phosphorylated by Aurora A on the spindle, which leads to the release of merotelically-attached, polymerizing MTs (Figure 6). Even when both correct and erroneous attachments are destabilized on the leading KT, the higher density of MT from the closer spindle pole will facilitate the formation of correct attachments. In contrast to Hec1-S55, the level of Hec1-S69 phosphorylation in metaphase is not related to the level of CIN. However, Hec1-S69 phosphorylation is also important for suppressing formation of lagging chromosomes [182], corroborating the relationship between chromosome oscillation, Hec1 phosphorylation, and error correction. How different phosphorylation sites on Hec1 tail are differentially regulated is currently unknown.

When the Aurora A activity gradient was integrated in a numerical model of chromosome oscillation, reduction of erroneous KT-MT attachments was reproduced [178]. In this model, an Aurora A-like activity gradient was considered in a force-balance model describing chromosome kinetics. The gradient peaks at spindle poles and declines toward the spindle equator, and the activity reduces Hec1 affinity to MT. In the simulation, the number of merotelic attachments sharply declined in the first few rounds of oscillatory KT motion. The numerical model also reproduced KT oscillatory motion, which is promoted by the Aurora A activity gradient that reduces KT-MT affinity when KTs approach spindle poles. Importantly, both the amplitude of chromosome oscillation and the efficiency of error correction are reduced not only when the Aurora A activity was suppressed, but also when it was upregulated. This is because high Aurora A activity close to the equator confines the range of KT motion that hampers selective destabilization of erroneous attachments. These simulation results may be relevant to the finding that Aurora A is generally upregulated in cancer [193].

## 5. Conclusions and Outlook

Regarding physiological roles of chromosome oscillation, several possibilities have been proposed, such as checking the correct balance of force across KTs as a “self-test” for error-free anaphase and prevention of entanglement or damage of chromosomes [158,176]. A recent report suggests another possibility that chromosome oscillation contributes to the correction of erroneous KT-MT attachments [147].

The error correction largely occurs in prometaphase, when MTs and KTs encounter stochastically, mainly through Aurora B-mediated Hec1 phosphorylation on KTs not under tension. However, it was shown that KT reorientation, which reflects the KT detachment and reattachment to MTs, occurs even in metaphase [51]. The previously-unappreciated role of chromosome oscillation in the correction of erroneous KT-MT attachments may ensure the establishment of bi-orientation as a final check before anaphase onset (Figure 8). Error correction by chromosome oscillation was also reported in yeast meiosis [194,195]. Even when merotelic attachments remain at anaphase onset and result in the formation of lagging chromosomes, several mechanisms work to resolve erroneous attachments during this phase ([113], *bioRxiv* doi:10.1101/2021.03.30.436326), ensuring mitotic fidelity.

Defective regulation of KT-MT attachments is manifested as abnormal chromosome dynamics, such as lagging chromosomes and chromosome misalignment. On the other hand, abnormal chromosome dynamics can cause defects in KT-MT attachment regulation, exemplified by the finding that attenuated chromosome oscillation reduces the efficiency of error correction of KT-MT attachments [147]. Another example is that delayed chromosome alignment increases the rate of chromosome missegregation, due to an increase in erroneous KT-MT attachment formation during delayed chromosome alignment and/or insufficient error correction during relatively shortened metaphase [196]. These relationships between chromosome dynamics and KT-MT attachment regulation were identified by direct observation of living cells, but not through genomic analysis or gene expression profiles, warranting microscopic study for investigating CIN.

The concept that attenuated chromosome oscillation is a cause of CIN has to be corroborated by specifying the underlying cause of attenuated chromosome oscillation in CIN cancer cells. It is also important to reveal how chromosome oscillation is dampened in the process of oncogenic transformation. Currently, mechanisms ensuring mitotic fidelity are mainly studied in cultured cancer cell lines, which have different properties from primary cancer cells growing in three-dimensional microenvironment [197,198]. Therefore, chromosome oscillation has to be observed in primary cancer cells under conditions similar to physiological circumstances, e.g., organoid culture [199,200]. Regarding Kif18A, the relationship between its roles in spindle assembly and chromosome oscillation needs to be clarified. It has been suggested that Kif18A activity must be kept in a proper range, because not only Kif18A depletion, but also Kif18A overexpression causes multipolar spindle formation [201]. Chromosome oscillation must also be kept in a proper range, as its hyperenhancement by Kif18A depletion was reported to cause KT detachment and micronuclei formation [189,202]. This is of clinical relevance, because Kif18A depletion specifically compromises survival of CIN cancer cells [188,189,190]. Whether lowering CIN level by Kif18A depletion through enhancing chromosome oscillation acts synergistically with spindle disruption for cancer therapy is an interesting possibility to be examined.

Defects in chromosome dynamics specifically seen in cancer cells can be a target for cancer therapy. Inhibitors for various mitotic motor proteins that alter chromosome dynamics, such as Eg5, CENP-E, and Kif18A, are now under investigation for the efficacy against cancer cells [203,204,205]. Aurora A and B are often dysregulated in cancer cells, and various Aurora kinase inhibitors are in clinical trials [193,206,207]. Hec1 is also a promising target for cancer therapy, and several inhibitors have been developed [208]. Further study on the relationship between chromosome dynamics and mitotic fidelity will pave the way for development of novel anti-cancer drugs.

## Figures and Tables

**Figure 1 cancers-13-04531-f001:**
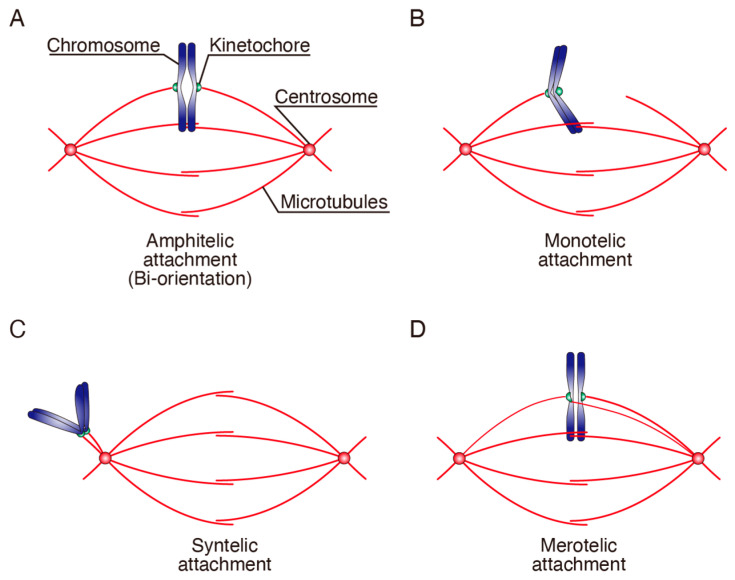
Correct and erroneous KT-MT attachments: (**A**) amphitelic attachment (bi-orientation), (**B**) monotelic attachment, (**C**) syntelic attachment, (**D**) merotelic attachment. See text for details.

**Figure 2 cancers-13-04531-f002:**
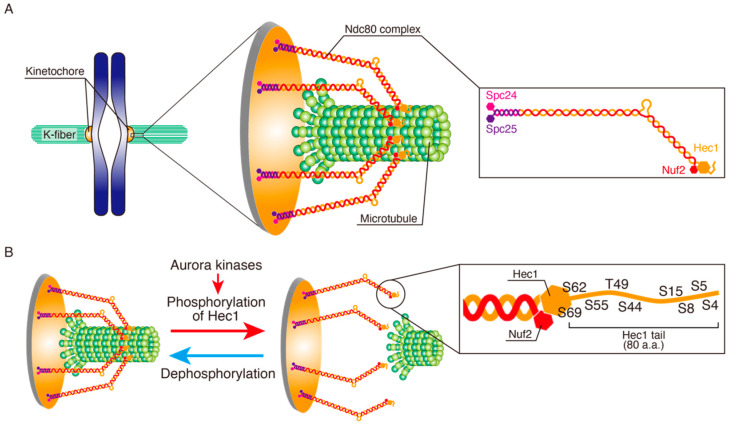
Regulation of KT-MT attachment through Hec1 tail phosphorylation by Aurora kinases. (**A**) On KTs in mammalian cells forming bi-orientation, multiple copies of the Ndc80 complex, composed of Hec1, Nuf2, Spc24, and Spc25, bind to the lateral surface of MTs to tether KTs to MT ends. (**B**) Phosphoregulation of KT-MT attachment. Phosphorylation of Hec1 tail by Aurora kinases reduces its affinity to MTs, allowing KT detachment, while dephosphorylation stabilizes the attachment. Phosphorylation sites on the Hec1 tail are shown in an inset.

**Figure 3 cancers-13-04531-f003:**
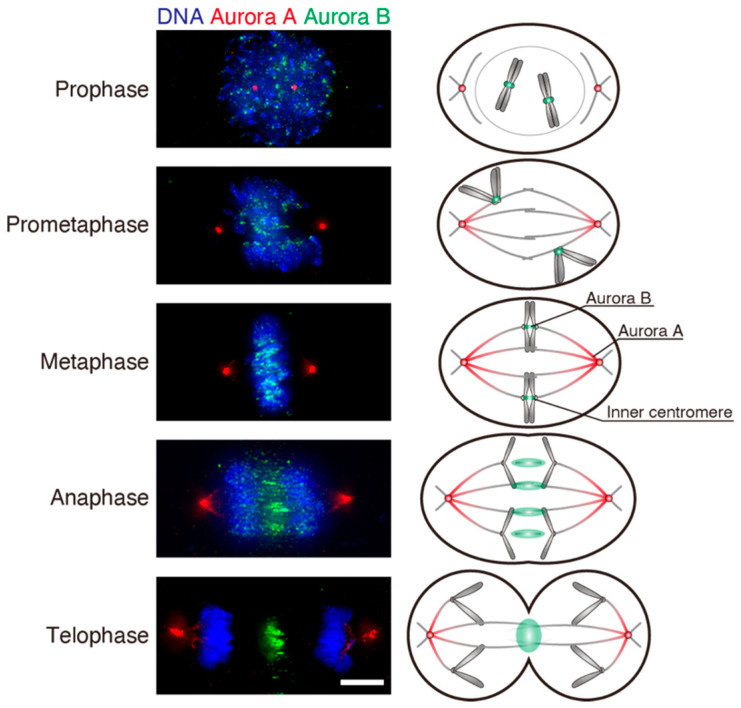
Localization of Aurora A and Aurora B during mitosis. Left: Immunofluorescence imaging of HeLa cells in different mitotic phases. HeLa cells were fixed with methanol and stained with Aurora A (red; Abcam, ab12875, 1:2000) and Aurora B (green; BD Bioscience, 611082, 1:2000) antibodies. DNA was stained with DAPI (blue). Scale bar: 5 μm. Right: Schematic diagrams showing localization of Aurora A (red) and Aurora B (green) in different mitotic phases.

**Figure 4 cancers-13-04531-f004:**
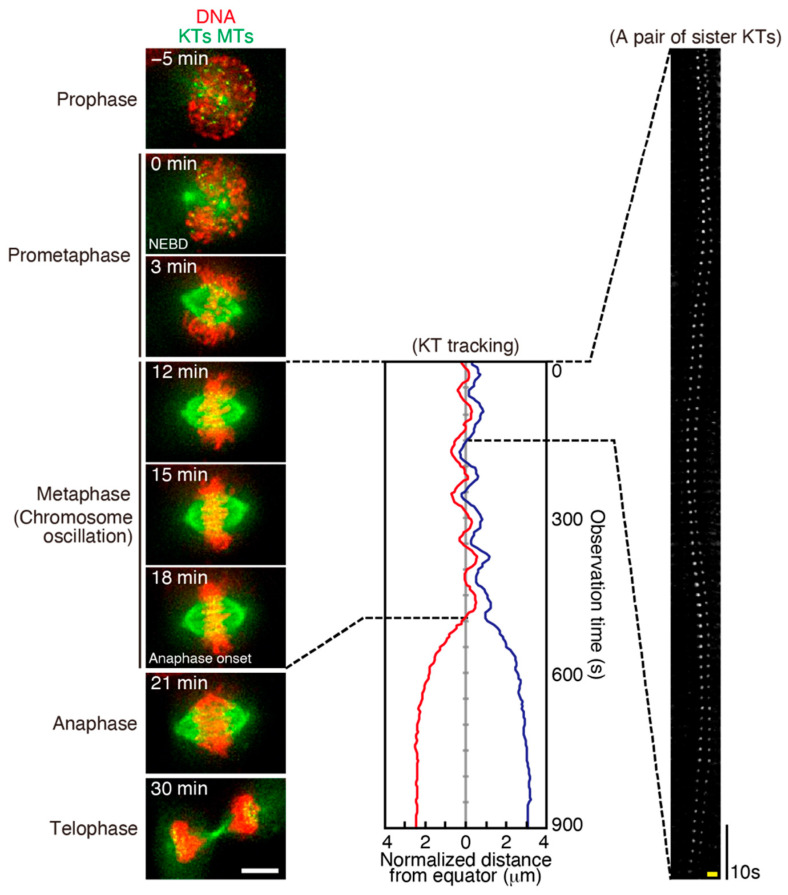
Chromosome oscillation. Left: Live cell imaging of an RPE-1 cell expressing EGFP-α-tubulin and EGFP-CENP-A, which visualize MTs (green) and KTs (green), respectively. Chromosomes (red) were visualized using SiR-DNA (Cytoskeleton, Inc., Denver, CO, USA, 200 nM). Images were collected every minute by a DeltaVision microscope (Cytiva). Representative images of different mitotic phases are shown. Times from nuclear envelope breakdown (NEBD) are indicated. Scale bar: 5 μm. Middle: Trajectories of a pair of sister KTs in an RPE-1 cell from metaphase to telophase plotted as the distance from the spindle equator. Right: A kymograph of the sister KTs in metaphase visualized by EGFP-CENP-A. Images were collected every 2 s by a DeltaVision microscope. Horizontal scale bar: 1 μm, vertical scale bar: 10 s.

**Figure 5 cancers-13-04531-f005:**
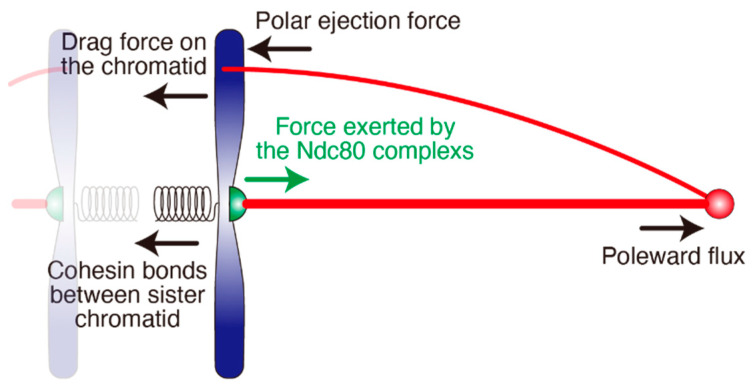
Forces acting on a sister chromatid. See text for details.

**Figure 6 cancers-13-04531-f006:**
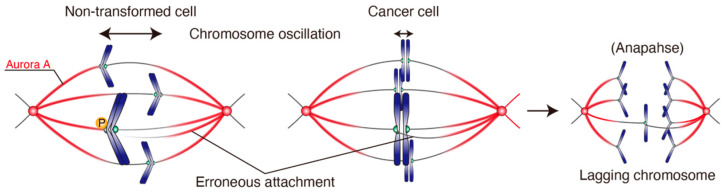
Chromosome oscillation facilitates Hec1 phosphorylation and error correction. In non-transformed cells, chromosome oscillation facilitates the Hec1-S55 phosphorylation by Aurora A, promoting the correction of erroneous KT–MT attachments. In cancer cells, chromosome oscillation is attenuated, which leads to reduced Hec1-S55 phosphorylation by Aurora A, resulting in inefficient error correction and increase in chromosome missegregation, such as the appearance of lagging chromosomes. P: phosphorylation.

**Figure 7 cancers-13-04531-f007:**
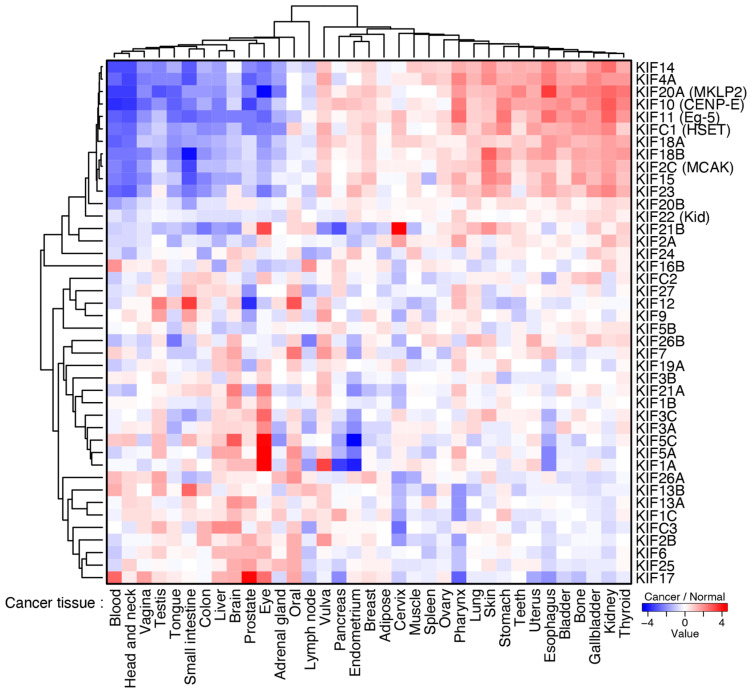
Expression of motor proteins in cancer. Expression datasets of kinesin superfamily for normal and cancer tissues were obtained from microarray gene expression data using GPL570 platform (HG-U133_Plus_2) in GENT2. GENT2: a platform for exploring Gene Expression patterns across Normal and Tumor tissues. Available online: http://gent2.appex.kr/gent2/ (accessed on 28 July 2021). The expression value of each kinesin gene in each tissue was averaged and the expression ratio of the cancer tissue to the normal tissue was calculated. Expression ratios were converted to binary logarithms and used for heatmapping and clustering, which were drawn by the heatmap.2 function of the gplots package in R. gplots: Various R Programming Tools for Plotting Data. Available online: https://CRAN.R-project.org/package=gplots (accessed on 4 August 2021). The R Project for Statistical Computing. Available online: https://www.R-project.org/ (accessed on 4 August 2021).

**Figure 8 cancers-13-04531-f008:**
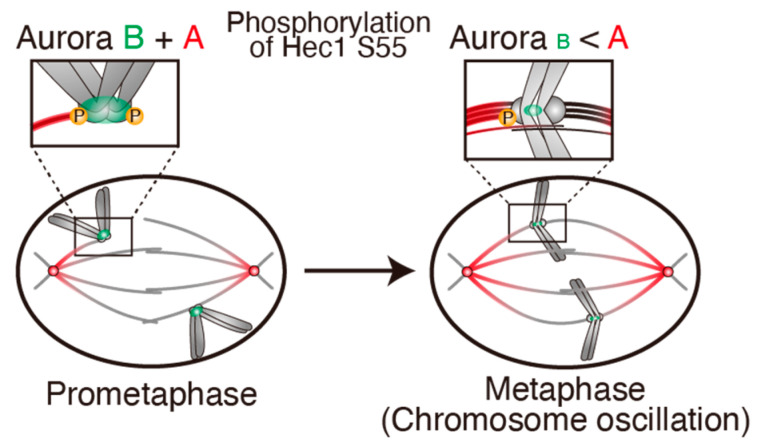
Error correction by Aurora B and Aurora A. During prometaphase, Hec1 on KTs is phosphorylated by Aurora B (green), which resides at the inner centromere, as well as by Aurora A (red), which localizes around spindle poles, facilitating correction of erroneous KT–MT attachments. In metaphase, Hec1 phosphorylation by Aurora B is reduced, while Aurora A on the spindle phosphorylates Hec1-S55 when KTs approach spindle poles through chromosome oscillation, thereby correcting any remaining erroneous attachments. P: phosphorylation.

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
