# Peer review of "Attenuated Chromosome Oscillation as a Cause of Chromosomal Instability in Cancer Cells"

_cancers, 2021, doi:10.3390/cancers13184531_

Round 1

Reviewer 1 Report

In this manuscript, Iemura et al summarized the current understanding of the establishment process of chromosome attachments to the spindle in mammalian cells with emphasis on the relationships between chromosome oscillation and correction of erroneous attachments and described a link between attenuated chromosome oscillation and chromosome instability in cancer cells. This is a well-organized review, providing a novel insight into aneuploidy generation in cancer cells. The manuscript is certainly of interest to general readers. I highly recommend the manuscript to be published in Cancers. I have only minor comments for improvement of the manuscript.

Minor comments:

  1. Figure 2

It may not be easy for general readers to understand the structure and phosphorylation sites of the Ndc80 complex without explanation or an additional illustration. Hec1, Nuf2, Spc25, and Spc25 in the complex are perhaps unclear for readers. In (B), explanation of phosphorylation sites present at Hec1 tail is probably required.

  1. Figures 3 and 4

If these results are original, procedures should be described. If they are not, their sources should be described.

  1. Figure 6

Aurora A gradient in illustration may be unclear.

  1. Line 390

Figure 6 is probably Figure 8.

Author Response

We thank the Reviewer for the encouraging comments.

  1. Figure 2: It may not be easy for general readers to understand the structure and phosphorylation sites of the Ndc80 complex without explanation or an additional illustration. Hec1, Nuf2, Spc25, and Spc25 in the complex are perhaps unclear for readers. In (B), explanation of phosphorylation sites present at Hec1 tail is probably required.

In response to the Reviewer’s comments, we explained the structure of the Ndc80 complex in the illustration and legend of Figure 2A. We also explained the phosphorylation sites at the Hec1 tail in the legend of Figure 2B.

  1. Figures 3 and 4: If these results are original, procedures should be described. If they are not, their sources should be described.

These results are original, and we described experimental procedures in the legends.

  1. Figure 6: Aurora A gradient in illustration may be unclear.

According to the Reviewer’s comment, we emphasized the Aurora A gradient in the illustration.

  1. Line 390: Figure 6 is probably Figure 8.

We confirmed that Figure 6 is correct as it shows dysfunction of error correction through chromosome oscillation in cancer cell lines, while Figure 8 shows error correction by Aurora A and B in prometaphase and metaphase.

Reviewer 2 Report

The review provides a very nice and detailed description on the causes and consequences of chromosome instability and most importantly highlights the much unappreciated and understudied role of chromosome oscillations in mitotic fidelity. As such, it also brings together recently published findings by the authors, that Hec1 plays a major role in this process. All figures are appropriate, nicely illustrated and help summarise the complexity of KT-MT attachment dynamics.

There are a few minor comments regarding grammar and clarification of some complex sentences (please see attached pdf file) that can easily be addressed.

In addition to this, I think that the authors need to provide additional explanations in some areas as listed below:

  1. Would be beneficial to have a brief section (either at the start or following section 1) on chromosome dynamics and evolutionary conservation (e.g. in plants, yeast).
  2. Provide a summary of what Hec1 actually is.....i.e. how it was originally identified as a Rb binding protein that is highly expressed in some cancers. This will provide added context to its critical role in mitotic maintenance and hence the focus on its regulation.
  3. Evidence of aberrant Hec1 levels in different cancers and targetting potential, including development of drugs (e.g. TAI-1) and their reported efficacy in targetting cancer cells and/or known efficacy in helping maintain chromosome oscillations during mitosis?
  4. Use of AURKA and AURKB inhibitors in cancer and thoughts on how these influence correction of erroneous KT-MT attachments?

Many of these queries can simply be incorporated into other sections in a few brief sentences.

Th authors have done a fantastic job with discussing and describing such a complex and relatively understudied area of cancer biology and dynamics. Well done!

Author Response

We thank the Reviewer for encouraging comments and valuable suggestions.

There are a few minor comments regarding grammar and clarification of some complex sentences (please see attached pdf file) that can easily be addressed.

We greatly appreciate the detailed editing of the manuscript. We amended the sentences according to the Reviewer’s comments.

In addition to this, I think that the authors need to provide additional explanations in some areas as listed below:

  1. Would be beneficial to have a brief section (either at the start or following section 1) on chromosome dynamics and evolutionary conservation (e.g. in plants, yeast).

We added a section on chromosome dynamics in mitosis and its evolutionary conservation as section 2.1.

  1. Provide a summary of what Hec1 actually is.....i.e. how it was originally identified as a Rb binding protein that is highly expressed in some cancers. This will provide added context to its critical role in mitotic maintenance and hence the focus on its regulation.

We explained how Hec1 was identified and it is overexpressed in cancers, citing related papers (Line 130-132, Citation #65, 66).

  1. Evidence of aberrant Hec1 levels in different cancers and targetting potential, including development of drugs (e.g. TAI-1) and their reported efficacy in targetting cancer cells and/or known efficacy in helping maintain chromosome oscillations during mitosis?

We described that Hec1 is also a promising target for cancer therapy and several inhibitors have been developed (Line 506-507, Citation #208). How these inhibitors affect chromosome oscillation is currently unknown.

  1. Use of AURKA and AURKB inhibitors in cancer and thoughts on how these influence correction of erroneous KT-MT attachments?

We mentioned that Aurora A and B are often dysregulated in cancer cells, and various Aurora kinase inhibitors are in clinical trials (Line 505-506, Citation #193, 206, 207).

Reviewer 3 Report

This manuscript reviews recent findings related to chromosome oscillations and chromosome instability. Overall it looks good, but I have a few suggestions:

  1. Line 36-37. Aneuploidy usually refers to numerical aberration but not structural aberration.
  2. Line 37-38. What does "mainly" refer to? Are there other causes of aneuploidy? If so, this should be discussed.
  3. Line 50. The two citations are for chromosome congression, which is a part of chromosome dynamics but not all.
  4. Line 78-80. The authors may consider these two citations suggesting stabilization of monotelics due to geometry: PMID 32053104, 28536121.
  5. Line 94. A simulation work showing microtubule turnover for error correction may be referenced: PMID 26424798.
  6. Line 95-96 (and line 416-417.). The meaning of "stochastic" should be explained, a citation showing that KT-MT binding is stochastic would be helpful.
  7. Line 98-99. The turnover rates (both on- and off-rates) are more direct than affinity (the ratio of the two rates).
  8. Line 120-122. Citation 66 does not provide direct evidence for the spatial separation model and could be removed here. In addition, the authors could mention a recent paper (PMID: 33904910) showing the first direct evidence that Aurora B induces kinetochore-microtubule detachment in cells (as opposed to in vitro).
  9. Line 127-129. "KT detachment" and "destabilization of the KT-MT attachment" seem redundant. 
  10. Line 135. The citation for MT-bound Aurora B contributing to error correction is missing (PMID: 30737408).
  11. Line 163-164. Is there a citation supporting the idea that syntelics activate the SAC via detachment?
  12. Line 208-211. Citation 121 does not seem relevant to this sentence. A simulation (more recent than the other citations here) showing directional instability may be considered: PMID 26417109.
  13. Line 212-214. An observation showing suppressed anaphase oscillation could be included: PMID: 27829144.
  14. Line 224-225. What experimental system do these measurements come from, and how general is the result (how much variability between systems)?
  15. Line 238. Is this the correct reference, or another paper from Wan and Salmon.
  16. Line 242-244. A key citation showing the underlying mechanism of KIF18A plus-end accumulation should be included (PMID: 21884977).
  17. Line 364-366. It's unclear how suppression of microtubule dynamics helps chromosome incorporation into the spindle.
  18. Line 399-409. The cited simulation work should be explained in more detail, since this is a major theme of the review. What assumptions/parameters go into the simulation? Why does increasing and decreasing Aurora A activity have similar effects?
  19. Conclusion and Outlook. This section could benefit from more discussion of open questions and also ideas about how the postulated model (described in section 4.4) could be tested.

Author Response

We thank the Reviewer for critical reading and detailed comments.

  1. Line 36-37. Aneuploidy usually refers to numerical aberration but not structural aberration.

We explained that in contrast to typical aneuploidy (whole chromosome aneuploidy), which is gain or loss of entire chromosome, amplification or loss of parts of chromosomes is called structural (or segmental) aneuploidy, referring related citations (Line 37-38, Citation #3-5).

  1. Line 37-38. What does "mainly" refer to? Are there other causes of aneuploidy? If so, this should be discussed.

We clarified that aneuploidy is caused by chromosome missegregation during mitosis, which is derived not only from mitotic defects, but also from defects in interphase such as replication stress, by citing a related paper (Line 39-41, Citation #6).

  1. Line 50. The two citations are for chromosome congression, which is a part of chromosome dynamics but not all.

We cited papers describing chromosome dynamics other than chromosome congression, such as initial KT capture and transport via lateral attachment and KT rotation in the process of bi-orientation establishment (Line 56, Citation #22-24).

  1. Line 78-80. The authors may consider these two citations suggesting stabilization of monotelics due to geometry: PMID 32053104, 28536121.

We cited the first one (Line 106, Citation #50), but not the second one, as it is on the geometry of lateral and end-on attachment, but not on back-to-back geometry of sister KTs.

  1. Line 94. A simulation work showing microtubule turnover for error correction may be referenced: PMID 26424798.

We cited the work (Line 122, Citation #58).

  1. Line 95-96 (and line 416-417.). The meaning of "stochastic" should be explained, a citation showing that KT-MT binding is stochastic would be helpful.

We explained that MTs attach to KTs through the process of dynamic instability, which occurs in a stochastic manner. We cited the related papers (Line 122-123, Citation 40, 59-61).

  1. Line 98-99. The turnover rates (both on- and off-rates) are more direct than affinity (the ratio of the two rates).

We changed the terminology (Line 125-126).

  1. Line 120-122. Citation 66 does not provide direct evidence for the spatial separation model and could be removed here. In addition, the authors could mention a recent paper (PMID: 33904910) showing the first direct evidence that Aurora B induces kinetochore-microtubule detachment in cells (as opposed to in vitro).

We removed the citation 66 and mentioned the recent paper suggested by the Reviewer (Line 172, Citation #89).

  1. Line 127-129. "KT detachment" and "destabilization of the KT-MT attachment" seem redundant. 

We deleted the phrase “by destabilization of the KT-MT attachment” (Line 162-163).

  1. Line 135. The citation for MT-bound Aurora B contributing to error correction is missing (PMID: 30737408).

We cited the paper (Line 170, Citation #88).

  1. Line 163-164. Is there a citation supporting the idea that syntelics activate the SAC via detachment?

As it has not been directly shown, we deleted the description (Line 201-202).

  1. Line 208-211. Citation 121 does not seem relevant to this sentence. A simulation (more recent than the other citations here) showing directional instability may be considered: PMID 26417109.

We changed the citations (Line 244, Citation #141).

  1. Line 212-214. An observation showing suppressed anaphase oscillation could be included: PMID: 27829144.

We cited the paper (Line 247, Citation #144).

  1. Line 224-225. What experimental system do these measurements come from, and how general is the result (how much variability between systems)?

We specified that the measurements are on RPE-1 cells, referring the citation showing the data (Line 259-260, Citation #145-147).

  1. Line 238. Is this the correct reference, or another paper from Wan and Salmon.

We agreed and corrected the reference (Line 274, Citation #151).

  1. Line 242-244. A key citation showing the underlying mechanism of KIF18A plus-end accumulation should be included (PMID: 21884977).

We included the citation (Line 279, Citation #154).

  1. Line 364-366. It's unclear how suppression of microtubule dynamics helps chromosome incorporation into the spindle.

We cited papers suggesting that increased rates of spindle MT polymerization in CIN cells confer an enhanced dependence on the role of Kif18A in limiting MT growth for spindle assembly (Line 401-403, Citation #190, 191).

  1. Line 399-409. The cited simulation work should be explained in more detail, since this is a major theme of the review. What assumptions/parameters go into the simulation? Why does increasing and decreasing Aurora A activity have similar effects?

We explained that an Aurora A-like activity gradient, which peaks at spindle poles and declines toward the spindle equator that reduces Hec1 affinity to MT, was considered in a force-balance model describing chromosome kinetics (Line 437-40). We also explained that KT oscillatory motion was reproduced in the model by the Aurora A activity gradient that reduces KT-MT affinity when KTs approach spindle poles (Line 442-444). The amplitude of chromosome oscillation and the efficiency of error correction were reduced even when Aurora A was upregulated because high Aurora A activity close to the equator confines the range of KT motion that hampers selective destabilization of erroneous attachments (Line 446-448).

  1. Conclusion and Outlook. This section could benefit from more discussion of open questions and also ideas about how the postulated model (described in section 4.4) could be tested.

We presented open questions and discussed how to test the postulated model (Line 485-501).